# Evaluating the Effects of Sugar Shift^®^ Symbiotic on Microbiome Composition and LPS Regulation: A Double-Blind, Placebo-Controlled Study

**DOI:** 10.3390/microorganisms12122525

**Published:** 2024-12-07

**Authors:** Gissel García, Josanne Soto, Michael Netherland, Nur A. Hasan, Emilio Buchaca, Duniesky Martínez, Martha Carlin, Raúl de Jesus Cano

**Affiliations:** 1Pathology Department, Clinical Hospital “Hermanos Ameijeiras” (HHA), Calle San Lázaro No 701, Esq.a Belascoaín, Centro Habana, La Habana 10400, Cuba; gisselgarcia2805@gmail.com; 2Clinical Laboratory Department, Clinical Hospital “Hermanos Ameijeiras” (HHA), Calle San Lázaro No 701, Esq.a Belascoaín, Centro Habana, La Habana 10400, Cuba; josanne.soto@infomed.sld.cu; 3EzBiome, 704 Quince Orchard Rd, Gaithersburg, MD 20878, USAhasan@ezbiome.com (N.A.H.); 4Internal Medicine Department, Clinical Hospital “Hermanos Ameijeiras” (HHA), Calle San Lázaro No 701, Esq.a Belascoaín, Centro Habana, La Habana 10400, Cuba; ebuchaca@infomed.sld.cu; 5Research and Development Department, Center for Genetic Engineering and Biotechnology of Sancti Spíritus (CIGBSS), Circunvalante Norte S/N, Olivos 3, Apartado Postal 83, Sancti Spíritus 60200, Cuba; duniesky.martinez@cigb.edu.cu; 6The BioCollective, LLC, 4800 Dahlia Street, G8, Denver, CO 80216, USA; martha.carlin@thebiocollective.com; 7Biological Sciences Department, California Polytechnic State University, San Luis Obispo, CA 93407, USA

**Keywords:** diabetes, microbiome, probiotics, lipopolysaccharides, SCFA, clinical trials, Sugar Shift

## Abstract

(1) Background: This study evaluated the effects of BiotiQuest^®^ Sugar Shift^®^, a novel probiotic formulation, for its impact on gut microbiome composition and metabolic health in type 2 diabetes mellitus (T2D). T2D is characterized by chronic inflammation and gut microbiome imbalances, yet the therapeutic potential of targeted probiotics remains underexplored. (2) Methods: In a 12-week randomized, double-blind, placebo-controlled trial, 64 adults with T2D received either Sugar Shift or placebo capsules twice daily. Each dose provided 18 billion CFU of eight GRAS-certified bacterial strains and prebiotics. Clinical samples were analyzed for metabolic markers, and microbiome changes were assessed using 16S rRNA sequencing and metagenomics. (3) Results: Sugar Shift significantly reduced serum lipopolysaccharide (LPS) levels, improved insulin sensitivity (lower HOMA-IR scores), and increased short-chain fatty acid (SCFA)-producing genera, including Bifidobacterium, *Faecalibacterium, Fusicatenibacter,* and *Roseburia*. Pro-inflammatory taxa like *Enterobacteriaceae* decreased, with reduced LPS biosynthesis genes and increased SCFA production genes. The *Lachnospiraceae:Enterobactericeae* ratio emerged as a biomarker of reduced inflammation. (4) Conclusions: These findings demonstrate the potential of Sugar Shift to restore gut homeostasis, reduce inflammation, and improve metabolic health in T2D. Further studies are warranted to explore its long-term efficacy and broader application in metabolic disease management.

## 1. Introduction

Diabetes mellitus is a global health concern characterized by impaired glucose metabolism [1,2]. In 2018, an estimated 537 million people were affected by T2D globally, with projections suggesting that this number could rise to 643 million by 2030 and 783 million by 2045 [3]. This alarming increase has led the scientific community to recognize diabetes as a pandemic affecting nations across all levels of development [4]. Current T2D therapies fail to address underlying microbiome imbalances contributing to inflammation and insulin resistance.

Type 2 diabetes mellitus (T2D), the most common form, is diagnosed when glucose intolerance is present (blood glucose ≥ 11.1 mmol/L) and fasting glucose levels are elevated (≥7 mmol/L or 126 mg/dL) [5]. Effective management generally requires lifestyle changes, oral antidiabetic agents like metformin, and maintaining glycosylated hemoglobin (HbA1c) levels below 7%. However, many patients struggle to achieve optimal glycemic control despite these interventions [4].

The development of T2D is influenced by a combination of genetic and environmental factors [6], with diets high in carbohydrates and processed foods contributing to microbiome imbalances. These imbalances often involve a reduction in beneficial bacteria and an overgrowth of bacteria that increase lipopolysaccharide (LPS) production, an endotoxin associated with inflammation and insulin resistance [7]. Chronic inflammation is now recognized as a critical factor in the progression of metabolic disorders such as T2D, with LPS from Gram-negative gut bacteria acting as a potent driver of this inflammation [8,9,10]. Elevated LPS levels can disrupt metabolic homeostasis and impair gut barrier integrity by activating pro-inflammatory pathways [11]. Certain bacterial families, including *Enterobacteriaceae, Bacteroidaceae,* and *Desulfovibrionaceae*, are known to produce LPS and are commonly found in higher abundance in individuals with metabolic syndrome and related inflammatory conditions [9,10,12].

The gut microbiota plays a central role in regulating metabolic and immune responses, influencing both glucose metabolism and inflammation [8,13]. Among the most studied phyla, Verrucomicrobia, particularly *Akkermansia muciniphila*, has garnered significant attention due to its role in improving gut barrier function, reducing inflammation, and mitigating metabolic diseases, including T2D [14,15,16,17]. However, T2D is also associated with the overrepresentation of lipopolysaccharide (LPS)-producing bacteria, such as *Enterobacteriaceae*, *Bacteroidaceae*, and *Desulfovibrionaceae*. These bacteria drive inflammation through LPS-induced pro-inflammatory pathways, disrupting metabolic homeostasis and exacerbating insulin resistance [7,10,11]. As a result, microbiome-targeted therapies have emerged as promising strategies to counteract these effects, with probiotics in particular showing potential to remodel the gut microbiome, improve glucose control, and enhance insulin sensitivity in individuals with diabetes [18,19].

BiotiQuest Sugar Shift [20] is a specially designed microbial consortium of eight generally-recognized-as-safe (GRAS) bacterial strains formulated using a community-based flux balance analysis [21] to enhance its impact on glucose and fructose metabolism, promoting their conversion to mannitol [18,20]. This unique formulation optimizes SCFA production and includes strains selected for their ability to reduce inflammation and improve gut health. In a 12-week, double-blind, placebo-controlled study involving 64 individuals with type 2 diabetes (T2D), Sugar Shiftdemonstrated significant efficacy in stabilizing fasting glucose, lowering serum LPS levels, and improving insulin sensitivity as measured by the HOMA-IR index [10]. 

In a 12-week, double-blind, placebo-controlled study of 64 individuals with T2D, Sugar Shift demonstrated efficacy in stabilizing fasting glucose, lowering serum LPS levels, and reducing insulin resistance as measured by the HOMA-IR index [18]. These findings suggest that Sugar Shift supports the growth of beneficial microbes, reduces LPS biosynthesis, and promotes SCFA production, contributing to improvements in metabolic and inflammatory markers in individuals with T2D.

This study builds on these findings by evaluating the effects of Sugar Shift on the gut microbiome and its metabolic outcomes, using both 16S rRNA sequencing and shotgun metagenomics. Sugar Shift enhances energy metabolism and gut barrier integrity and reduces inflammation by fostering the growth of SCFA-producing bacteria while reducing LPS-associated taxa. This dual action mitigates inflammation, stabilizes HbA1c, improves insulin sensitivity, and supports overall metabolic health.

## 2. Materials and Methods

### 2.1. Ethical Considerations

A comprehensive human safety and efficacy assessment of BiotiQuest Sugar Shift symbiotic formulation was conducted in a randomized, double-blind, placebo-controlled clinical trial to evaluate its probiotic potential in healthy adults. The study protocol was approved by the Hermanos Ameijeiras Clinical and Surgical Hospital Ethics Committee under approval number CEI-HHA-0001, dated 22 June 2021, the National Institute of Nutrition of Cuba, and the Cuban Ministry of Health, adhering to good clinical practice protocols and the Declaration of Helsinki [22]. This trial was registered with the Cuban Public Registry of Clinical Trials (RPCEC) under the registration number RPCEC00000413, dated 6 December 2022; and the ISRCTN registry number ISRCTN48974890.

### 2.2. Study Design

A 12-week clinical study was conducted at the Hermanos Ameijeiras Clinical and Surgical Hospital (HHA) in La Habana, Cuba, to investigate the effect of BiotiQuest Sugar Shift (SS) as a supplementary therapeutic approach for T2D. The study was randomized, double-blind, and placebo-controlled. Patients were enrolled from June 2021 to April 2022. 

The sample size was calculated using a two-sided alpha of 0.05, a 95% confidence level, a standardized mean difference of 0.75, and a test power of 80%, with an expected loss proportion of 10%. Based on these parameters, the minimum required sample size was determined to be 28 individuals per group. To account for potential losses, the sample size was adjusted to 32 participants per group. Ultimately, 64 patients consented to participate in the study.

The inclusion criteria for the study consisted of participants aged 30 to 65 years (inclusive), of any gender or skin color, with a clinical diagnosis of T2D and who were able and willing to provide informed written consent. Exclusion criteria included patients with chronic kidney disease, oncoproliferative diseases, pregnant women, and individuals with intellectual or psychological dysfunction that impaired their ability to understand and comply with the study requirements, as determined by the principal investigator. Patients meeting the inclusion criteria were enrolled by an attending physician. The study enrolled 64 patients and randomly assigned them to either the SS cohort or the placebo group. The SS cohort consisted of 32 patients receiving the test substance (SS), while the placebo group (P) consisted of 32 patients who were given a placebo. Table 1 provides an overview of the demographic and other baseline characteristics of the study population, and Figure 1 summarizes the study in a CONSORT diagram [23].

Patients were randomly assigned to treatment groups using EpiData 3.1 software [26]. The software generated a list of random numbers for each group, which was used to assign patients to their respective groups based on their compliance with the inclusion and signature criteria for informed consent. To ensure blinding, the products were dispensed in identical packaging.

Patients aged 30 to 65 years, diagnosed with type 2 diabetes (T2D) [27], with a baseline body mass index (BMI) between 18.5 and 40, and attending specialized diabetes consultations at the Hermanos Ameijeiras Hospital were eligible for the study. Participation was open to all sexes and skin colors, and provided informed consent was given. Exclusion criteria included chronic kidney disease, oncoproliferative diseases, and pregnancy.

The “test substance”, Sugar Shift, was manufactured by Blister Pack Pro, LLC (Lafayette, CO, USA). Each capsule contains 96 mg (18 billion CFU) of a bacterial consortium of eight strains of GRAS-classified bacteria that include *Bacillus subtilis* De111™, *Bifidobacterium bifidum*, *Bifidobacterium longum*, *Lactobacillus paracasei*, *Lactobacillus plantarum* TBC0036, *Lactobacillus reuteri*, *Leuconostoc mesenteroides* TBC0037, and *Pediococcus acidilactici.* Each capsule contains 370 mg of prebiotics and fillers consisting of inulin, microcrystalline cellulose, D-mannitol, and stearic acid. The placebo capsules, indistinguishable in appearance from the test substance, were similarly manufactured by Blister Pack Pro, LLC (Lafayette, CO, USA), and contained the same ingredients in identical proportions as the test substance sans the microbial consortium. Each placebo capsule contained 50 mg each of the prebiotics inulin and D-mannitol and 270 mg of the fillers microcrystalline cellulose and stearic acid.

Subjects consumed two Sugar Shift or placebo capsules per day, 12 h apart. Two capsules provided daily an optimal concentration of beneficial strains based on prior studies. The foil packaging containing the capsules were distributed every 28 days over the 12-week study period after blood sample collection for clinical determination.

### 2.3. Sample Collection

Clinical samples were collected and processed as previously described [18]. All clinical determinations were conducted using a Cobas 6000 modular immunochemical autoanalyzer (Roche Diagnostics, Indianapolis, IN, USA), a fully automated system designed for high-throughput and precise biochemical and immunochemical analyses. Each clinical parameter was measured in accordance with Roche Diagnostics’s protocol, which included reagent preparation, instrument calibration, sample handling, and analysis procedures. 

Fecal samples from ten patients from each cohort group were collected on Day 1 and Day 84 of the study, using DNA/RNA Shield™ Fecal Collection Tubes (Zymo Research, Irvine, CA, USA) to ensure the preservation and stabilization of microbial DNA and RNA, following the manufacturer’s guidelines [28]. Samples were then stored in a secure, dry environment until further processing for microbiome characterization. A total of 40 fecal samples were preserved in this manner for consistency and to maintain the integrity of the nucleic acids for downstream analysis.

### 2.4. Lipopolysaccharide (LPS) Determinations

The serum LPS levels of participating patients were measured using the ToxinSensor™ Endotoxin Detection System (Version 11242021) from GenScript (GenScript USA Inc., 860 Centennial Ave, Piscataway, NJ 08854, USA). This chromogenic assay detects endotoxin (lipopolysaccharide, LPS) levels in biological samples, expressed in endotoxin units (EUs). The assay was conducted following the manufacturer’s instructions to ensure accuracy and reliability.

Briefly, serum samples were diluted as necessary to fall within the detectable range of the assay, and all reagents were prepared and calibrated according to the protocol provided by GenScript. Each sample was added to the ToxinSensor™ 96-well plate, along with appropriate positive and negative controls to validate the assay’s performance. Following the addition of the ToxinSensor™ chromogenic substrate, the reaction was allowed to develop for the recommended time, after which absorbance was measured at 545 nm using a microplate reader. LPS concentrations were calculated based on the standard curve generated with known endotoxin standards, and the results were expressed in EU/mL.

All samples were processed in duplicate to confirm reproducibility, and strict aseptic techniques were followed to avoid contamination. This method allowed for the sensitive detection of LPS levels, providing reliable data for the assessment of endotoxemia in each cohort.

### 2.5. HOMA-IR Index Calculation

The homeostatic model assessment of insulin resistance (*HOMA-IR*) index was calculated to assess insulin resistance in the study participants. Fasting glucose and insulin levels were measured, and the HOMA-IR score was calculated using the following formula:$$HOMA\text{-}IR=\frac{Fasting\;Insulin\left(µ\frac{U}{\mathrm{mL}}\right)\times Fasting\;GLucose(\frac{\mathrm{mg}}{\mathrm{dL}})}{405}$$

This formula is based on the original model developed by Matthews et al. [29] and is widely used as a simple method to estimate insulin resistance [29,30]. Fasting blood samples were collected after an overnight fast of at least 8 h to ensure accurate baseline insulin and glucose levels.

Insulin levels were determined using an immunoassay, and glucose concentrations were measured via an enzymatic assay on a Cobas 6000 analyzer (Roche Diagnostics, Indianapolis, IN, USA). All measurements were conducted in duplicate, and quality control samples were included to ensure the precision and accuracy of the assays. The HOMA-IR index was then calculated for each participant and used to assess the degree of insulin resistance.

### 2.6. Metagenome Analysis

#### 2.6.1. 16S rRNA Gene Metagenomic Sequencing

Metagenomics 16S sequencing was performed by EzBiome (Gaithersburg, MD, USA). DNA concentration was measured using the QuantiFluor dsDNA System on a Quantus Fluorometer (Promega, Madison, WI, USA). The 16S rRNA primers (V3–V4) within the ribosomal transcript were amplified using the primer pair containing the gene-specific sequences and Illumina adapter overhang nucleotide sequences. The primer sequences are: IlluminaF: CCTACGGGNGGCWGCAG; and IlluminaR: GACTACHVGGGTATCTAATCC.

Amplicon PCR processing was performed to amplify DNA templates from input samples. Each 25 µL of PCR reaction consisted of 12.5 ng of input DNA, 12.5 µL of 2× KAPA HiFi HotStart ReadyMix (Kapa Biosystems, Wilmington, MA, USA), and 5 µL of 1 µM of each primer. The PCR cycling conditions included an initial denaturation at 95 °C for 3 min, followed by 25 cycles of denaturation at 95 °C for 30 s, annealing at 55 °C for 30 s, and extension at 72 °C for 30 s, with a final elongation step at 72 °C for 5 min. The PCR products were purified using Mag-Bind RxnPure Plus Magnetic Beads (Omega Biotek, Norcross, GA, USA).

A second PCR amplification was performed to add barcodes and sequencing adapters to the final PCR product. Each 25 µL of reaction used the same master mix conditions as previously described. The cycling conditions were as follows: an initial denaturation at 95 °C for 3 min, followed by 8 cycles of denaturation at 95 °C for 30 s, annealing at 55 °C for 30 s, and extension at 72 °C for 30 s, with a final elongation step at 72 °C for 5 min.

The libraries were normalized using the Mag-Bind^®^ EquiPure Library Normalization Kit (Omega Biotek, Norcross, GA, USA) and subsequently pooled. The pooled library was evaluated for quality using an Agilent 2200 TapeStation and sequenced with a 2 × 300 bp paired-end read setting on the MiSeq platform (Illumina, San Diego, CA, USA).

#### 2.6.2. 16S Metagenomic Taxonomic Profiling

For paired-end sequencing, two sequences, representing each end of the same PCR amplicon, were merged to obtain the overlapping sequence information using the VSEARCH program v 2.29.1 [31]. Primers were trimmed and removed using an in-house code, and then sequences were quality-filtered to remove low-quality reads and reads not predicted to be 16S reads. The VSEARCH program was used to search the EzBioCloud 16S database (23 August 2023) [32] and calculate the sequence similarities of the query NGS reads. A cutoff of 97% 16S similarity was used for the species-level identification. Species not matched by 97% percent were then clustered using UCLUST v 11.0.667 [33] tool with a 97% similarity boundary, where a group of clusters was defined as an OTU. Using the OTU information (number of OTUs and sequences in each OTU), various alpha diversity measures can be calculated, including species richness and Shannon and Simpson diversity indices.

#### 2.6.3. Metagenomic Shotgun Sequencing

Metagenomic shotgun sequencing was conducted by EzBiome (Gaithersburg, MD, USA). The concentration of genomic DNA (gDNA) was measured using the QuantiFluor dsDNA System on a Quantus Fluorometer (Promega, Madison, WI, USA). For library construction, 50 ng of gDNA was processed with the Kapa HyperPlus Kit (Kapa Biosystems, Wilmington, MA, USA).

Briefly, gDNA was enzymatically sheared, and the DNA fragments underwent end repair, 3′ adenylation, and adapter ligation according to the manufacturer’s instructions. Adapter-ligated libraries were then PCR-amplified using the following protocol: an initial denaturation at 98 °C for 45 s, followed by 5 cycles of denaturation at 98 °C for 15 s, annealing at 60 °C for 30 s, and extension at 72 °C for 30 s, with a final elongation step at 72 °C for 1 min. PCR products were purified using Mag-Bind RxnPure Plus Magnetic Beads (Omega Biotek, Norcross, GA, USA).

The resulting libraries were quantified and assessed for quality using D1000 ScreenTape on an Agilent 2200 TapeStation (Agilent Technologies, Inc., Santa Clara, California, USA. Libraries were normalized, pooled, and sequenced on an Illumina HiSeq X10 platform (Illumina, San Diego, CA, USA) with a paired-end 150 bp run configuration.

#### 2.6.4. Metagenomic Functional Profiling

For each sample, functional annotations were derived by aligning reads to the KEGG database, v 112.0 [34] using DIAMOND, v 2.1.0 [35]. DIAMOND was executed using the blastx parameter, which translates each metagenomic read into six open reading frame variations and matches the resulting amino acid sequences against the pre-built KEGG database. In cases where a read matched multiple KEGG entries, the top hit was selected. Once the KEGG orthologs were quantified, MinPath, v 1.6 [36] was utilized to infer the presence of KEGG functional pathways.

#### 2.6.5. Metagenomic Taxonomic Profiling

The profiling process began by assessing the potential presence of bacterial and archaeal species in each raw metagenomic sample read by using Kraken2 v 2.1.3 [37] and a pre-built core gene database [38] containing k-mers (k = 35) of reference genomes obtained from the EzBioCloud database [39]. Fungi and viral full genomes from NCBI’s RefSeq (https://www.ncbi.nlm.nih.gov/refseq/) (accessed on 3 November 2024) were also added to the Kraken2 database to enhance taxonomic resolution.

Based on the list of candidate species identified, a custom Bowtie2 v 2.5.4 [40] database was built utilizing the core genes and genomes from the species found during the first step. Raw metagenomic reads were then aligned to this Bowtie2 database using the “--very-sensitive” option and a quality threshold of phred33. The resulting BAM file was converted and sorted using Samtools v 1.21 [41]. The coverage of the mapped reads against the BAM file was calculated using Bedtools v 2.31.1 [42].

To minimize false positives, an in-house script was employed to quantify the reads mapping to a species only when the total coverage of its core genes (for archaea and bacteria) or genome (for fungi and viruses) exceeded 25%. The species abundance was then determined based on the total number of reads mapped, and normalized species abundance was calculated by dividing by the total length of all references for the respective species.

Taxonomic biomarkers were also explored using LEfSe v 1.1.2 [43], which utilizes a linear discriminant analysis (LDA) effect size method to support high-dimensional class comparisons. LEfSe’s biomarkers were found using a *p*-value < 0.05 (Kruskal–Wallis test) [44] and an LDA score (log 10) > 2.0.

#### 2.6.6. Alpha Diversity Assessment

To compare alpha diversity across the baseline, placebo, and treated groups at the end of the study (day 84), we utilized a suite of diversity metrics that measure species richness, diversity, and phylogenetic relationships. Richness-based metrics included the abundance-based coverage estimator (ACE) [45], which emphasizes rare species, and Chao1 [46], which estimates undetected species based on low-abundance observations. Both metrics were calculated using the *fossil* package in R software [47].

Observed OTUs [48] were also calculated as a direct measure of species richness within each sample. Diversity indices, including the Shannon Index [49] and its non-parametric variant (NPShannon) [49], were used to evaluate both species richness and evenness, while the Simpson Index quantified the likelihood of two randomly selected individuals belonging to different species, with lower values indicating higher diversity. These were determined using the vegan package in R [50].

To assess phylogenetic diversity, we calculated the phylogenetic diversity (PD) [51] metric, which measures the total branch length of a phylogenetic tree encompassing all observed species, providing insight into evolutionary relationships within the microbiome. All diversity metrics were calculated using rarefied data to ensure consistent sequencing depth across samples. Statistical analyses, including ANOVA [52] or Kruskal–Wallis [44] tests, were conducted to identify differences in alpha diversity among the groups, with post hoc tests being applied to determine specific group differences where appropriate. A significance level of *p* < 0.05 was used for all comparisons.

## 3. Results

### 3.1. Clinical Results

The clinical data revealed significant improvements in key metabolic markers following Sugar Shift treatment. The intervention group demonstrated a marked reduction in serum lipopolysaccharide (LPS) levels and improved insulin sensitivity, as indicated by lower homeostatic model assessment of insulin resistance (HOMA-IR) scores (Figure 2). These changes align with the observed shifts in gut microbiome composition, including an increase in SCFA-producing bacteria. While the study did not directly measure gut barrier integrity, the reduction in serum LPS levels suggests a potential decrease in endotoxin translocation, a known driver of systemic inflammation and metabolic dysfunction. These findings underscore the connection between microbiome modulation, reduced inflammatory burden, and improved metabolic outcomes in type 2 diabetes.

The figures were generated using the ggpubr package in R software, presenting the distributions of LPS and HOMA-IR values with jittered data points for clarity. Statistical significance across groups was determined using the Kruskal–Wallis test [44], with significant differences observed between the Sugar Shift-treated group and both the placebo control and baseline values of both cohorts at time zero. Error bars represent the interquartile range, and individual points reflect raw data values.

### 3.2. Metagenome Quality

The 16S rRNA gene sequencing yielded an average of 21,324 ± 7281 raw reads per sample. After quality filtering, which involved trimming low-quality bases and removing chimeric sequences, an average of 19,170 ± 6546 high-quality reads with Phred values ≥ 30 per sample were retained, accounting for 89.9% of the total raw reads. Rarefaction curves demonstrated sufficient sequencing depth, with saturation being observed in all samples, indicating comprehensive coverage of the microbial communities. Alpha diversity metrics showed no significant impact of sequencing depth on richness or diversity indices, supporting the adequacy of read counts across samples.

Shotgun metagenomic sequencing generated an average of 17,563,173 ± 5,302,580 raw reads per sample. Quality control procedures, including trimming adapters, filtering low-quality bases (Phred < 30), and discarding short reads (<50 bp), retained 96.9% of the raw reads, yielding an average of 17,023,173 ± 5,153,619 high-quality reads per sample. The average GC content across samples was 46.98%, aligning with expected microbial genomic profiles. Coverage analysis showed uniform sequencing depth across all samples, ensuring robust comparative analyses.

### 3.3. Changes in Alpha Diversity

The ACE index [53] results, summarized in Figure 3, indicate a significant difference in species richness between the baseline and the Sugar Shift-treated group at Day 84, with a *p*-value of 0.041. There is no significant difference in species richness between the placebo group at Day 84 and the baseline (*p* = 0.6). Additionally, there is a significant difference between the Sugar Shift-treated and placebo groups at Day 84, with a *p*-value of 0.02.

The Chao1 [46] index results, summarized in Figure 3, indicate a significant difference in species richness between the baseline and the Sugar Shift-treated group at Day 84, with a *p*-value of 0.00097. There is no significant difference in species richness between the placebo group at Day 84 and the baseline (*p* = 0.67). Additionally, there is a significant difference between the Sugar Shift-treated and placebo groups at Day 84, with a *p*-value of 0.022.

The Shannon diversity index [49] results, summarized in Figure 3, indicate no significant difference in species richness between the baseline and the Sugar Shift-treated group at Day 84, with a *p*-value of 0.13. There is also no significant difference in species richness between the placebo group at Day 84 and the baseline (*p* = 0.73). Additionally, there is no significant difference between the Sugar Shift-treated and placebo groups at Day 84, with a *p*-value of 0.45.

### 3.4. Functional Annotation and Pathway Analysis of Metagenomic Data

Shotgun sequencing data were used to evaluate the functional potential of the gut microbiome, focusing on genes associated with lipopolysaccharide (LPS) biosynthesis and short-chain fatty acid (SCFA) production. The abundance of genes involved in LPS biosynthesis were quantified and compared between the baseline and the end of the study (day 84) for the Sugar Shift-treated cohort. Generally, there were significant decreases in LPS biosynthetic genes in the Sugar Shift-treated cohort as compared with both the baseline and the placebo group. Conversely, four genes linked to SCFA production were analyzed and showed significant increases in abundance when compared with the baseline and placebo cohorts. The results are summarized in Table 2, presenting the statistical comparisons of gene abundances between the two time points for the treated group.

LEfSe [43] analysis using PICRUSt v 2 2.5.3 [55] revealed significant changes in the functional potential of the gut microbiome in the treated group compared with the baseline. Specifically, the relative abundance of the K14138 ortholog, encoding acetyl-CoA synthase, was significantly increased (*p* = 0.022), suggesting enhanced microbial metabolic capacity for acetyl-CoA production. Furthermore, the K13013 ortholog, which is involved in O-antigen biosynthesis, also showed a significant increase in abundance in the treated group (*p* = 0.034).

### 3.5. Taxonomic Composition and Findings

LEfSe analysis of the 16S metagenome, normalized for gene copy number, identified several discriminatory biomarkers with significantly different abundances among the baseline, placebo, and treated groups at the end of the study (day 84). These biomarkers represent taxa that differentiate the microbial communities across the groups and provide insights into the effects of the treatment. The results of this analysis, including the identified biomarkers and their respective statistical significance, are presented in Table 3. The table includes the taxa with significantly different abundances among baseline, placebo, and treated groups at the end of the study (day 84), along with the corresponding *p*-values. Biomarkers were determined to be significant at a threshold of *p* < 0.05.

Taxonomic analysis of shotgun metagenomic data at the family level revealed significant differences in the relative abundances (RAs) of key bacterial families between baseline, placebo, and Sugar Shift-treated cohorts at the end of the study (Day 84). Among the families analyzed, *Enterobacteriaceae* exhibited a significant decline in RA in the Sugar Shift-treated group compared with the baseline group (*p* < 0.001), while *Lachnospiraceae* showed a significant increase in RA in the treated group (*p* < 0.001). *Bacteroidaceae* displayed a slight, but not statistically significant, decrease in RA for the Sugar Shift group. Similarly, *Bifidobacteriaceae* showed an increase in RA in the treated group, though this change was not significant (*p* > 0.5). These results are illustrated in Figure 4.

A more detailed taxonomic analysis at the genus level revealed dynamic changes (flux) in specific genera between the baseline and Sugar Shift-treated cohort. These genera, which are key contributors to the observed family-level shifts and may have a potential impact on the functional changes reported in Table 2, are illustrated in Figure 5, emphasizing the microbial dynamics within the gut microbiome following treatment. This figure illustrates the changes in relative abundance (RA) of 12 key genera in the gut microbiome, comparing baseline microbiomes with those from the placebo and Sugar Shift-treated cohorts at the end of the study (Day 84). The genera shown were selected based on their significant or notable shifts in RA, contributing to the family-level changes observed in Figure 4. The boxplots represent the distribution of RA for each genus, with smaller black dots indicating individual samples (jittered for visualization) and larger black dots denoting the mean RA for each group. Statistical significance was assessed using the Kruskal–Wallis test [44].

Among the 12 genera analyzed, notable increases in RA were observed in the Sugar Shift-treated cohort at the end of the study compared with both the placebo and baseline groups. Specifically, significant increases were detected in *Bifidobacterium*, a genus associated with probiotic properties; *Fusicatenibacter*, an emerging genus linked to gut health; Faecalibacterium, a butyrate-producing genus critical for maintaining gut barrier integrity; and Roseburia, another butyrate-producing genus known for its beneficial effects on gut health. These changes highlight key microbial dynamics associated with the treatment and align with the family-level shifts observed in Figure 4.

Figure 6 compares the ratios of Bacillota to Bacteroidota (FB) and *Lachnospiraceae* to *Enterobacteriaceae* (LE) to evaluate their potential as biomarkers for inflammation in type 2 diabetes (T2D). The Bacillota to Bacteroidota ratio showed no significant differences among the cohorts, indicating limited utility as a discriminatory biomarker in this context. In contrast, the LE ratio demonstrated significant differences between the treated group and baseline (*p* = 0.011) and between the treated group and the placebo group (*p* = 0.052). No significant difference was observed between the baseline and placebo groups (*p* = 0.68). These results suggest that the LE ratio is a more sensitive and specific marker for evaluating microbial changes related to inflammation in T2D, particularly in response to the probiotic intervention.

## 4. Discussion

This study builds on previously published clinical findings of the symbiotic Sugar Shift (SS) in the management of type 2 diabetes (T2D) [18], emphasizing its microbiome-mediated impact on insulin resistance and serum lipopolysaccharides (LPS), key drivers of metabolic dysfunction in T2D. By integrating clinical and microbiome data, we highlight how targeted microbial modulation can improve metabolic and inflammatory outcomes in T2D patients.

### 4.1. Clinical Parameters

Lipopolysaccharides (LPSs), structural components of Gram-negative bacterial outer membranes, play a central role in driving inflammation and immune dysregulation in chronic diseases like T2D [57]. Increased intestinal permeability allows LPS to enter systemic circulation, where it activates toll-like receptor 4 (TLR4) in immune cells, triggering the release of pro-inflammatory cytokines such as TNF-α, IL-6, and IL-1β [58]. This low-grade inflammation, termed metabolic endotoxemia, disrupts insulin signaling pathways, exacerbates insulin resistance, and accelerates the progression of T2D [59]. Chronic exposure to elevated LPS levels can overwhelm the immune system, inducing immune exhaustion and further impairing the regulation of inflammation. These findings underscore the critical role of LPS in perpetuating the inflammatory–metabolic cycle of T2D. Targeting LPS production or translocation via microbiome modulation is, therefore, a promising therapeutic strategy.

In this study, the SS symbiotic significantly reduced serum LPS levels, reflecting improved gut barrier function and reduced intestinal inflammation. These results align with prior studies suggesting that microbiome-targeted interventions can mitigate gut permeability and systemic inflammatory responses [60,61,62]. The observed reduction in LPS in the SS-treated group also correlates with improved insulin resistance, as measured by the homeostatic model assessment of insulin resistance (HOMA-IR) [30]. These findings provide direct evidence of how gut microbiota modulation can influence metabolic endotoxemia and insulin signaling.

Short-chain fatty acids (SCFAs), such as acetate, propionate, and butyrate, also emerged as critical mediators in this study. SCFAs, known for their role in promoting gut barrier integrity and modulating inflammation, have consistently been linked to metabolic health [58]. In line with these observations, the SS intervention was associated with increased SCFA levels, which likely contributed to the stabilization of clinical parameters such as fasting glucose and HbA1c. Unlike many previous studies that focus on general populations, this study used a targeted symbiotic intervention, revealing significant changes in SCFA levels within just 12 weeks. The findings further demonstrated a clear link between increased SCFA production and the reduction in LPS, underscoring the synergistic effects of these microbial metabolites on metabolic health [63].

Unique to this study is the simultaneous evaluation of SCFA production, LPS reduction, and clinical outcomes, particularly in a diabetic cohort. This comprehensive approach provides insights that are directly translatable to clinical settings. By addressing both microbial dysbiosis and metabolic dysfunction, SS offers a novel therapeutic strategy for T2D management, emphasizing the interplay between gut microbiota, inflammation, and insulin sensitivity. These results align with and expand upon prior research, demonstrating that reducing metabolic endotoxemia can improve glycemic control and insulin sensitivity in metabolic disorders [64].

The integration of these findings with previously published clinical data highlights SS as a promising intervention for managing T2D. By reducing gut inflammation, improving intestinal barrier integrity, and mitigating endotoxemia, SS directly addresses key drivers of metabolic dysfunction. Future studies should focus on elucidating the mechanistic pathways linking microbiome changes to long-term metabolic outcomes. Additionally, longitudinal studies are needed to determine the durability of these effects and to explore SS’s potential applications in precision medicine for other metabolic and inflammatory disorders.

### 4.2. Metagenomic Sequences

The sequencing data generated in this study demonstrate high quality and depth, providing a reliable foundation for subsequent microbiome analyses. The rarefaction curve analysis confirmed that sequencing depth was sufficient to capture the diversity within the samples, as saturation was observed across all datasets. Importantly, alpha diversity metrics indicated no significant impact of sequencing depth on richness or diversity indices, validating the adequacy of the sequencing effort and minimizing potential biases [65,66].

The shotgun metagenomic sequencing results further corroborate the quality and reliability of the data. These results highlight the reliability of the data for investigating microbial community composition and functional potential. The combination of 16S rRNA gene sequencing and shotgun metagenomics provides complementary insights, with the former capturing taxonomic diversity and the latter offering detailed functional resolution. This dual approach enables a holistic understanding of microbial dynamics within the studied samples [67].

Furthermore, the high-quality sequencing data lay the groundwork for robust statistical analyses, minimizing noise and enhancing the interpretability of the findings. The uniformity of sequencing depth and the retention of a high proportion of reads in both approaches reflect the effectiveness of the experimental protocols and the suitability of the data for downstream applications, such as diversity analysis, taxonomic profiling, and functional annotation [68].

### 4.3. Alpha Diversity Trends

Alpha diversity, a measure of species richness and evenness in microbial communities, is a key indicator of gut microbiome health [69,70]. In this study, differences in alpha diversity among baseline, placebo, and Sugar Shift (SS)-treated groups were assessed using the ACE, Chao1, and Shannon diversity indices, with statistical significance being evaluated by the Kruskal–Wallis test.

Increased richness reflects a healthier, more resilient gut microbiome, with greater variety and proliferation of rare, keystone taxa involved in essential metabolic and ecological functions. These changes likely contribute to enhanced metabolic flexibility, improved gut barrier function, and increased short-chain fatty acid (SCFA) production, such as butyrate, which supports intestinal health and reduces inflammation [71].

The increase in diversity holds important implications for gut health. A diverse microbiome is associated with enhanced metabolic flexibility, improved gut barrier function, and greater resilience against dysbiosis and pathogenic invasion [72,73]. Specifically, the expansion of microbial taxa observed with SS treatment may contribute to increased production of short-chain fatty acids (SCFAs) such as butyrate, which are vital for maintaining intestinal health, reducing inflammation, and regulating metabolic processes [74]. These functional enhancements align with the clinical improvements observed in serum lipopolysaccharide (LPS) levels and insulin resistance, as discussed elsewhere in this study.

Overall, these findings underscore SS’s ability to enhance microbial richness, promoting a diverse and functionally robust microbiome, which aligns with observed clinical improvements in LPS levels and insulin resistance. Further research should explore the functional implications of these changes, particularly SCFA production and immune modulation.

### 4.4. Functional Analysis Trends

The functional potential of the gut microbiome was evaluated using shotgun sequencing to investigate changes in genes associated with LPS biosynthesis and short-chain fatty acid production. The Sugar Shift intervention demonstrated a significant impact on these functional pathways, based on metagenomic data, with notable shifts in gene abundances that reflect the intervention’s role in modulating the microbial ecosystem.

A significant reduction in the abundance of genes involved in LPS biosynthesis was observed (Table 2) in the SS-treated cohort compared with both baseline and placebo groups. LPS, a major component of the outer membrane of Gram-negative bacteria, is a key driver of metabolic endotoxemia and systemic inflammation, which are often associated with insulin resistance and chronic metabolic conditions [75]. The reduction in LPS biosynthetic genes suggests that the SS intervention may attenuate the pro-inflammatory potential of the gut microbiome, aligning with the observed clinical improvements in serum LPS levels and insulin resistance. These findings support the hypothesis that targeted microbiome modulation can reduce inflammation-related metabolic dysregulation by altering microbial gene expression [76].

In contrast, genes associated with SCFA production demonstrated significant increases in abundance in the SS-treated cohort (Table 2) compared with both baseline and placebo groups. SCFAs, including butyrate, acetate, and propionate, are critical metabolites produced by gut bacteria through the fermentation of dietary fibers [77]. They play a central role in maintaining gut barrier integrity, modulating immune responses, and regulating host metabolism [78]. The increased abundance of SCFA-related genes suggests that the SS intervention fosters a microbiome capable of enhanced metabolic activity, potentially contributing to improved gut and systemic health.

The LEfSe analysis using PICRUSt revealed functional shifts in the gut microbiome. Notably, the relative abundance of the K14138 ortholog, which encodes acetyl-CoA synthase, was significantly higher in the SS-treated group (*p* = 0.022). Acetyl-CoA, a crucial metabolic intermediate, plays a central role in various biosynthetic and energy-producing pathways, including the production of short-chain fatty acids (SCFAs). This increase suggests an enhanced microbial capacity for acetyl-CoA synthesis, potentially contributing to the observed upregulation of SCFA-related genes and their associated health benefits.

The K13013 ortholog, linked to O-antigen biosynthesis, was significantly more abundant in the SS-treated group (*p* = 0.034). O-antigens are structural components of lipopolysaccharides (LPSs), and changes in their biosynthesis may reflect alterations in microbial community composition or LPS structural variability. This finding suggests a potential reconfiguration of the gut microbial community toward a less pro-inflammatory state, supported by the observed decrease in overall LPS biosynthetic gene abundance.

After 84 days of treatment with the SS formulation, a significant increase in the activity of the acetyl-CoA pathway (*p* = 0.017) was observed in the LEfSe analysis from the PICRUSt data, indicating enhanced microbial utilization of CO_2_ in the gut [79,80]. This upregulation reflects an elevated capacity for carbon fixation by acetogenic bacteria, promoting the production of short-chain fatty acids (SCFAs), particularly acetate [81]. Probable contributing taxa include members of the *Lachnospiraceae* family, such as *Blautia* and *Ruminococcus*, which are known for their acetogenic and SCFA-producing capabilities [82], and *Ruminococcaceae*, including *Faecalibacterium prausnitzii*, a key butyrate and acetate producer. Additionally, taxa in the *Bifidobacteriaceae* family (e.g., *Bifidobacterium*) play a complementary role in acetate production and metabolism [83]. These microbial families collectively enhance acetate production and its conversion to acetyl-CoA, supporting colonocyte energy production, gut barrier integrity, and anti-inflammatory effects. This metabolic shift underscores the potential of the probiotic formulation to modulate gut microbial functions and improve systemic health, particularly in the context of inflammation and insulin resistance associated with type 2 diabetes.

These functional changes underscore the dual impact of the Sugar Shift (SS) intervention in reducing pro-inflammatory microbial potential while enhancing beneficial metabolic pathways. Specifically, the observed reductions in genes associated with lipopolysaccharide (LPS) biosynthesis and the concomitant increases in short-chain fatty acid (SCFA)-related genes highlight the therapeutic potential of SS in addressing gut dysbiosis and mitigating its downstream effects on metabolic health. Future studies should aim to link these functional alterations to specific microbial taxa and explore their direct influence on host physiology. Furthermore, longitudinal research is necessary to evaluate the durability of these effects and their implications for long-term health outcomes. Collectively, these findings reinforce the potential of SS as a targeted strategy to improve gut microbiome functionality and systemic metabolic health.

In contrast, metformin-induced alterations in the gut microbiota of patients with T2D have been associated with a reduction in microbial alpha diversity, although beta diversity remained relatively stable throughout treatment. Many participants in this study were concurrently using metformin for T2D management. These observations suggest that incorporating SS into the treatment regimen for T2D may help counterbalance the negative impact of metformin on microbial diversity, supporting a more stable and beneficial gut microbial ecosystem in these patients [84].

### 4.5. Taxonomic Diversity Analysis 

Gut microbiota-derived LPS is an important factor involved in the onset and progression of inflammation and metabolic diseases, including T2D [85]. The composition of the gut microbiota is altered in T2D with a concomitant reduction of SCFA producers, notably butyrate [86]. LPSs, bacterial surface glycolipids produced by Gram-negative bacteria, are known to induce acute inflammatory reactions [87], particularly in the context of sepsis. However, LPSs can also trigger chronic inflammation [88]. In the case of T2D and other metabolic diseases originating from chronic inflammation of the gut, the source of LPSs is not an external infection, but rather an increase in endogenous production, which is usually sustained by the gut microbiota [87].

Taxonomic analysis of shotgun metagenomic data revealed significant differences in the relative abundances (RAs) of key bacterial families between the groups. Notably, *Enterobacteriaceae*, a family often associated with gut inflammation and dysbiosis [89], exhibited a significant decline in RA in the SS-treated group compared with the baseline group (*p* < 0.001). This reduction aligns with the reported decreases in lipopolysaccharide (LPS) biosynthetic genes, suggesting a shift away from pro-inflammatory microbial populations. In contrast, *Lachnospiraceae*, a family of obligate anaerobes known for their butyrate production and anti-inflammatory properties, showed a significant increase in RA in the SS-treated group (*p* < 0.001). This increase highlights the potential of SS to enrich beneficial taxa that support gut health and metabolic function.

The healthy intestine is typically characterized by a low-oxygen environment that supports large communities of obligate anaerobes, particularly from the phylum Bacillota [90,91,92,93]. Dysbiosis, commonly observed in conditions with underlying chronic inflammation, often involves a reduction in these obligate anaerobes and an increase in facultative anaerobes, particularly from the family *Enterobacteriaceae* [93]. This shift suggests a disruption in the gut’s anaerobic environment, with oxygen playing a key role in driving this imbalance [90,94]. Factors such as intestinal inflammation and antibiotic treatments elevate oxygen levels in the colon, disrupting anaerobiosis and fostering the growth of facultative anaerobic bacteria, including *Enterobacteriaceae*. This expansion of facultative anaerobes is not only a marker of dysbiosis but may also indicate epithelial dysfunction [95]. As oxygen availability in the large intestine rises, it provides a respiratory advantage to facultative anaerobes, leading to their dominance and further contributing to gut dysbiosis [96]. Understanding these dynamics opens potential diagnostic and therapeutic strategies, such as targeting oxygen levels to rebalance the microbiota, particularly in patients with conditions like IBD, where current treatments may be insufficient. These findings highlight the critical role of oxygen in maintaining a balanced gut microbiota and provide new insights into the interplay between host physiology, microbial community structure, and disease and the impact that Sugar Shift has on restoring this anaerobic environment. This is seen in the significant reduction in serum LPS (Figure 1) and the concomitant improvement in insulin resistance (Figure 1), largely a reflection of gut inflammation [97]. 

After 84 days of treatment with the Sugar Shift (SS) probiotic formulation, significant shifts in the gut microbiome were observed, with implications for anti-inflammatory properties, the promotion of an anaerobic environment, and the reduction in the relative abundance (RA) of Enterobacteriaceae. The LEfSe analysis of the PICRUSt data revealed a significant increase in the activity of the acetyl-CoA pathway (*p* = 0.017), indicating enhanced microbial utilization of CO_2_ in the gut. This upregulation reflects elevated acetogenic activity, particularly from taxa belonging to the families *Lachnospiraceae, Ruminococcaceae,* and *Bifidobacteriaceae* [98]. These taxa are key producers of short-chain fatty acids (SCFAs) like butyrate and acetate, metabolites critical for maintaining gut health and reducing inflammation.

The enriched taxa promote an anaerobic gut environment by producing SCFAs that stimulate oxygen consumption by colonocytes, thereby maintaining the oxygen-free conditions necessary for the proliferation of obligate anaerobes. This metabolic activity simultaneously suppresses the growth of facultative anaerobes, which thrive in oxygen-rich conditions often associated with gut inflammation. The reduction in *Enterobacteriaceae* RA following SS treatment is consistent with a healthier gut microbial balance, as this family is a major producer of pro-inflammatory lipopolysaccharides (LPSs) linked to metabolic endotoxemia and insulin resistance [89,99]. The suppression of *Enterobacteriaceae* is congruent with the observed functional and clinical improvements, including reduced serum LPS levels and improved insulin sensitivity (Figure 2).

There was a significant increase in the relative abundance of *Bifidobacteriaceae* between both pre-treatment groups and the Sugar Shift group after 12 weeks of treatment. Increased relative abundances of this genus is notably reported in association with successful treatment with probiotics and in non-diabetic, “healthy” individuals [100,101]. It is also plausible to hypothesize that the Bifidobacteria present in Sugar Shift colonize the intestines and help restore the homeostasis of the gut microbiota and increase the relative abundance of SCFA producers, thus alleviating inflammation. This interaction, however, would need to be supported by whole-genome and metabolomic studies.

The observed increase in *Bifidobacterium*, particularly *Bifidobacterium adolescentis* (*p* = 0.048) in the Sugar Shift-treated group compared with both baseline and placebo, highlights the treatment’s selective impact on promoting beneficial gut microbes. This genus is well established for its probiotic potential, contributing to gut health through mechanisms such as the production of short-chain fatty acids (SCFAs) [102], competitive exclusion of pathogens [103], and modulation of the immune system [104].

The specific increase in *B. adolescentis* is particularly significant, as this species is associated with fiber metabolism and the production of acetate and lactate, which serve as precursors for butyrate synthesis by other beneficial microbes [105,106]. This increase likely contributes to the functional ecosystem shifts observed in the gut microbiome, enhancing overall microbial diversity and metabolic function.

The genus-level increase in *Bifidobacterium* aligns with the observed family-level enrichment in *Bifidobacteriaceae*, suggesting a coordinated response at different taxonomic levels. This correlation underscores the consistency of the treatment’s effects across microbiome scales and highlights the potential for Sugar Shift to support gut health through the targeted proliferation of beneficial bacterial taxa. These results emphasize the importance of *Bifidobacterium* in the gut microbial community and its pivotal role in driving the beneficial effects associated with the intervention.

It is noteworthy that *Faecalibacterium* is highly represented in the gut microbe of Sugar Shift-treated patients. This taxon is generally abundant in the microbiome of “healthy” individuals, but it is present at reduced levels in individuals with gastrointestinal inflammatory diseases (e.g., diabetes). It has therefore been suggested to constitute a marker of a healthy gut and is associated with anti-inflammatory properties [107,108]. Among its attributes, *F. prausnitzii* is an acetate consumer that produces butyrate and bioactive anti-inflammatory molecules such as shikimic and salicylic acids [109]. The significant decrease in serum LPS noted among the Sugar Shift-treated study group may be attributed, at least in part, to the role of *F. prausnitzii* in the production of SSFAs and other anti-inflammatory substances.

*Fusicatenibacter* is an emerging genus of interest in gut health research, known for its potential anti-inflammatory properties and its role in maintaining intestinal homeostasis [110]. Members of this genus are closely associated with the fermentation of dietary fibers and the production of short-chain fatty acids (SCFAs), particularly butyrate, which is essential for maintaining gut barrier integrity and regulating immune responses. An increased abundance of *Fusicatenibacter* has been linked to healthier gut microbial profiles and reduced inflammation in certain conditions [111]. The observed increase in the treated group suggests that the intervention may foster the growth of beneficial butyrate-producing bacteria, contributing to improved gut health and functional outcomes. This aligns with previous findings indicating the genus’s positive correlation with metabolic and inflammatory markers.

Our study demonstrates that the symbiotic formulation Sugar Shift significantly reduces LPS concentrations, aligning with improvements in insulin resistance, lipid profiles, and fasting blood glucose (FBG) levels. Microbiome analysis supports these findings, highlighting the relationship between gut dysbiosis in T2D patients and elevated LPS levels, which contribute to increased inflammatory biomarkers in serum [112]. These results provide valuable insights into potential diagnostic and therapeutic strategies for managing T2D by targeting gut microbiome imbalances.

The results presented in Table 3 highlight significant shifts in the gut microbial community following Sugar Shift (SS) intervention, with several taxa being identified as discriminatory biomarkers across the baseline, placebo, and SS-treated groups at Day 84. Key taxa such as PAC001135_s, PAC001233_s, and *Alistipes finegoldii* showed significant increases in relative abundance in the SS-treated group compared with baseline (*p* = 0.03314, 0.02572, and 0.03909, respectively). These taxa, particularly *Alistipes finegoldii*, are associated with beneficial functions, including anti-inflammatory properties and short-chain fatty acid (SCFA) production [113,114], suggesting their role in improving gut health and metabolic regulation. Conversely, the baseline-specific taxon *Bacteroides eggerthii* has been identified as an inflammatory biomarker [115] and showed marked reductions in the SS-treated group, suggesting a shift toward a healthier microbial composition. In contrast, the placebo group demonstrated minimal changes in microbial composition, as taxa like PAC001458_s and PAC001458_g were only detected in this group, emphasizing the specific impact of SS treatment on gut microbiota. 

Collectively, these taxonomic shifts reflect a rebalancing of the microbial ecosystem in the SS-treated group, characterized by the enrichment of beneficial taxa and the depletion of potentially harmful ones. This rebalancing is consistent with the observed functional changes, including increased SCFA-related genes and reduced LPS biosynthesis, as well as clinical outcomes such as improved insulin resistance and reduced serum LPS levels. These findings underscore the targeted effects of the SS intervention in promoting a healthier gut microbiome, with implications for improved metabolic and inflammatory health.

The anti-inflammatory effects of the SS treatment are further supported by the increased abundance of butyrate-producing bacteria such as *Faecalibacterium* and *Roseburia*, which enhance gut barrier integrity by regulating tight junction proteins and preventing LPS translocation. These taxa also modulate immune responses, suppressing pro-inflammatory cytokines like TNF-α and IL-6 while promoting anti-inflammatory cytokines such as IL-10. The enrichment of *Alistipes* and *Coprococcus*, known for their roles in SCFA production and immune modulation, reinforces the anti-inflammatory potential of the SS intervention [116,117].

In parallel, the increase in acetogenic taxa such as *Blautia* and *Ruminococcus* further highlights the probiotic’s ability to enhance microbial metabolic capacity, favoring acetogenic pathways over competing methanogenesis. This shift promotes SCFA production, supports colonocyte energy production, and stabilizes the microbial ecosystem through cross-feeding interactions [118]. Overall, the SS intervention not only enriched beneficial SCFA-producing and anti-inflammatory taxa but also suppressed pro-inflammatory microbial populations, fostering a balanced gut microbiome. 

After 84 days of treatment with the probiotic formulation, a significant increase in the activity of the acetyl-CoA pathway (*p* = 0.017) was observed, indicating enhanced microbial utilization of CO_2_ in the gut. This upregulation suggests an elevated capacity for carbon fixation by acetogenic bacteria, likely contributing to increased production of beneficial short-chain fatty acids (SCFAs), such as acetate. The heightened activity of the pathway reflects a shift in microbial metabolic dynamics, favoring acetogenic processes over competing methanogenesis. These findings highlight the potential of the probiotic formulation to modulate gut microbial functions, improving carbon utilization and promoting SCFA-mediated benefits, such as gut barrier integrity and anti-inflammatory effects.

### 4.6. Biomarkers for Gut Health

The FB ratio is widely recognized as a biomarker for gut health, reflecting shifts in microbial composition and metabolic balance [119,120]. However, the LE ratio may provide a more precise and functionally relevant indicator of gut health. Unlike the FB ratio, which broadly examines changes at the phylum level, the LE ratio focuses on specific microbial families with well-defined roles in gut oxygenation and inflammation. *Lachnospiraceae*, obligate anaerobes, are key producers of short-chain fatty acids (SCFAs), which support gut barrier integrity, reduce inflammation, and promote anaerobic conditions by consuming oxygen in the gut epithelium [98,121]. In contrast, *Enterobacteriaceae*, facultative anaerobes, thrive in oxygen-rich environments typically associated with gut inflammation and dysbiosis [12]. Their proliferation is linked to lipopolysaccharide (LPS) production, metabolic endotoxemia, and chronic inflammation [122]. The LE ratio directly reflects the balance between these beneficial anaerobes and potentially harmful oxygen-tolerant bacteria. A higher LE ratio indicates a healthier gut environment with anaerobiosis and reduced inflammatory potential, while a lower LE ratio suggests dysbiosis and increased oxygenation.

As illustrated in Figure 6, significant differences were observed in the LE ratio but not the FB ratio among the study groups. While the FB ratio showed no significant changes across baseline, placebo, and treated groups, the LE ratio demonstrated significant increases in the treated group compared with both baseline (*p* = 0.011) and the placebo group (*p* = 0.0052), with no significant difference between baseline and placebo (*p* = 0.68). These findings suggest that the LE ratio is a more sensitive and specific biomarker for capturing microbial changes associated with inflammation and dysbiosis in T2D. Its ability to reflect shifts in gut oxygen dynamics and inflammatory potential highlights its utility in evaluating interventions, such as Sugar Shift, designed to restore microbial balance and improve gut health.

### 4.7. Mechanisms of Gut Microbiome Modulation by the Probiotic Sugar Shift in Type 2 Diabetes Management

Sugar Shift (SS) is a probiotic designed to enhance gut microbiome composition and function, delivering comprehensive gastrointestinal and systemic health benefits. It operates through several key mechanisms. First, SS promotes competitive exclusion [123], where formulation strains outcompete pathogenic bacteria for nutrients and adhesion sites. This process reduces the abundance of Gram-negative bacteria and their associated production of lipopolysaccharides (LPSs), a major driver of metabolic endotoxemia [124]. The decrease in LPS may also reflect improved gut barrier integrity, as SS strains enhance tight junction function, reducing intestinal permeability and systemic inflammation [125]. Additionally, SS stimulates the production of short-chain fatty acids (SCFAs), particularly butyrate, which strengthens the gut barrier, reduces inflammation, and provides energy to colonocytes [60]. By fostering SCFA-producing bacteria like *Faecalibacterium prausnitzii*, SS creates a positive feedback loop that supports anti-inflammatory microbial taxa [126]. It also enhances nutrient utilization by breaking down dietary fibers into fermentable substrates, fueling beneficial microbes and suppressing opportunistic pathogens.

The proprietary SS formula combines carefully selected strains to maximize synergistic benefits. *Bacillus subtilis* (DE111^®^) supports immune function, balances gut pH to inhibit pathogens, promotes beneficial microbes, and produces antimicrobial peptides [127,128]. *Bifidobacterium bifidum* aids butyrate production, digestion, and immune health, while *Bifidobacterium longum* enhances nutrient absorption and immune modulation [104,129]. *Lactobacillus plantarum* (TBC0036™) generates beneficial metabolites, including mannitol, sorbitol, and butyrate [130]. *Lactobacillus reuteri* converts glucose to mannitol, reduces inflammation, and enhances oxytocin production [131]. *Leuconostoc mesenteroides* (TBC0037™) produces mannitol, anti-inflammatory compounds, and vitamin B12, supporting overall gut health [132]. Finally, *Pediococcus acidilactici* (TBC0068™) produces pediocin A, which inhibits pathogens, prevents inflammation, and strengthens the gut lining [133,134]. Together, these synergistic strains comprise a guild that regulates glucose metabolism, reduces inflammation, and promotes gut health, positioning BiotiQuest Sugar Shift as a promising natural intervention for managing type 2 diabetes.

### 4.8. Limitation of the Study

Our study had several limitations that warrant consideration. First, the relatively small sample size (*n* = 27–30) and short duration may limit the generalizability of our findings and reduce the statistical power to detect subtle but potentially meaningful effects. These constraints emphasize the need for larger, multicenter studies with extended follow-up periods to validate our results and to assess the durability of the observed changes over time.

Second, while participants self-reported notable benefits during probiotic capsule consumption, these observations were not systematically captured. Participants consistently described improved stool regularity with softer stools, an absence of diarrhea associated with the oral treatment, a decreased sensation of hunger, and an overall improvement in well-being following the intervention. Although these anecdotal findings align with the intended benefits of the probiotic intervention, the absence of a standardized instrument to measure these outcomes introduces potential biases. For future studies, these subjective benefits will be formally evaluated through validated questionnaires to provide a more robust assessment of participants’ experiences.

Another limitation lies in the microbiome analysis. The number of samples available for microbiome profiling was limited, which may have affected our ability to detect significant changes in the relative abundance of certain bacterial families or genera. Expanding the sample size in future research will be critical to increase the power of microbiome analyses and to confirm the reported findings.

Finally, the duration of the intervention may have been insufficient to fully realize and measure the microbiome changes necessary for sustained clinical benefits. Although significant changes in microbial composition were observed after three months of treatment, longer durations, such as six months, may be required to optimize microbiome alterations and achieve meaningful improvements in metabolic health and glycemic control for patients with T2D.

These limitations underscore the need for future studies with larger sample sizes, longer follow-up periods, more comprehensive data collection methods, and additional metrics to evaluate both microbial and clinical outcomes. Addressing these gaps will enhance our understanding of the long-term efficacy and mechanisms of probiotic interventions in diabetes management.

## 5. Conclusions

Our findings demonstrate that the probiotic intervention induced significant shifts in microbiome composition, characterized by an increased *Lachnospiraceae:Enterobacteriaceae* (LE) ratio. This shift reflects a greater abundance of beneficial anaerobes, such as *Lachnospiraceae*, which are key producers of short-chain fatty acids (SCFAs) like butyrate and propionate, and a reduction in facultative anaerobes, such as *Enterobacteriaceae*, which are associated with lipopolysaccharide (LPS) production and inflammation. These compositional changes directly correlated with enhanced microbiome functionality, including increased expression of genes involved in SCFA production pathways and a decrease in genes linked to LPS biosynthesis. Together, these findings highlight the interplay between microbial composition and functional capacity, demonstrating how the probiotic intervention promotes a healthier, more balanced gut microbiome with reduced inflammatory potential.

The modulation of gut microbial composition and function by SS underscores the therapeutic potential of this symbiotic in promoting gut health, reducing systemic inflammation, and mitigating metabolic dysfunction, particularly in conditions such as T2D. The observed shifts in microbial dynamics, combined with functional enhancements, highlight the pivotal role of probiotics in restoring gut homeostasis and improving systemic metabolic health.

## Figures and Tables

**Figure 1 microorganisms-12-02525-f001:**
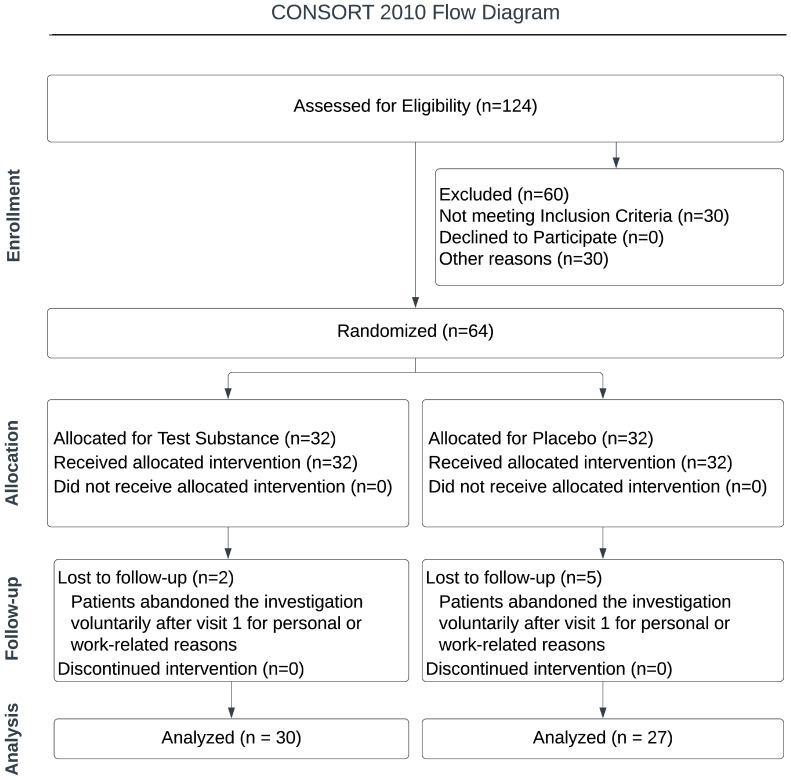
CONSORT [23] flow diagram of recruitment and retention throughout the study.

**Figure 2 microorganisms-12-02525-f002:**
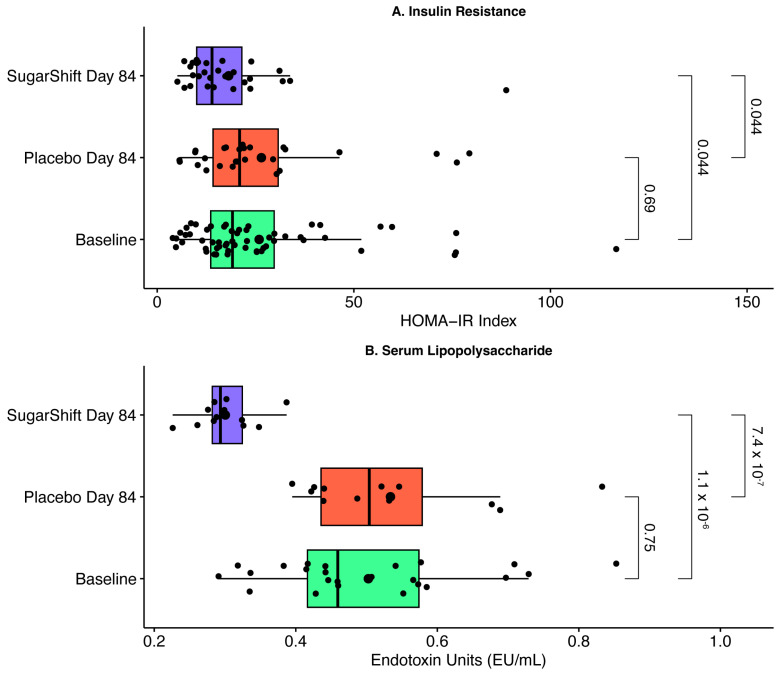
Serum lipopolysaccharide (LPS) levels and HOMA-IR values for the Sugar Shift-treated, placebo control, and baseline groups.

**Figure 3 microorganisms-12-02525-f003:**
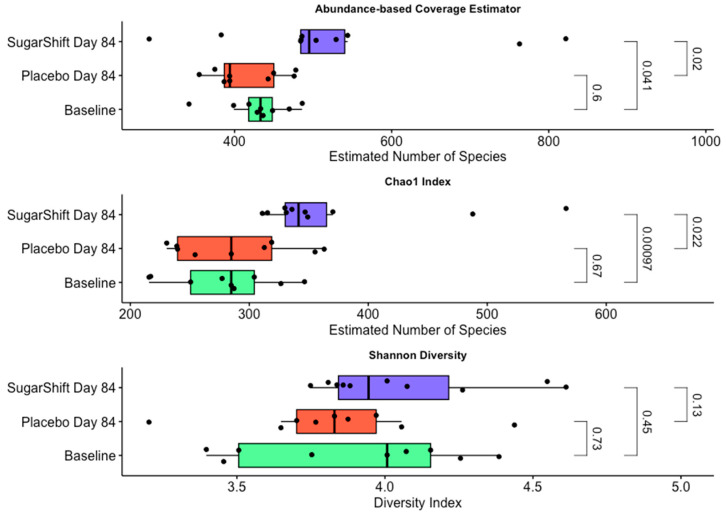
Comparison of alpha diversity indices across baseline, placebo, and Sugar Shift-treated groups. Boxplots with jitter representing the ACE, Chao1, and Shannon diversity indices for the baseline, placebo (Day 84), and Sugar Shift-treated (Day 84) groups. Data visualizations were generated using the ggpubr package in R software. Statistical comparisons were conducted using the Kruskal–Wallis test [44] to evaluate differences in species richness and diversity between groups. The Shannon index was calculated using the *vegan* v 2.6-8 package in R software [50] based on the relative abundance data of bacterial taxa identified, which included 23 phyla and 56 classes of bacteria.

**Figure 4 microorganisms-12-02525-f004:**
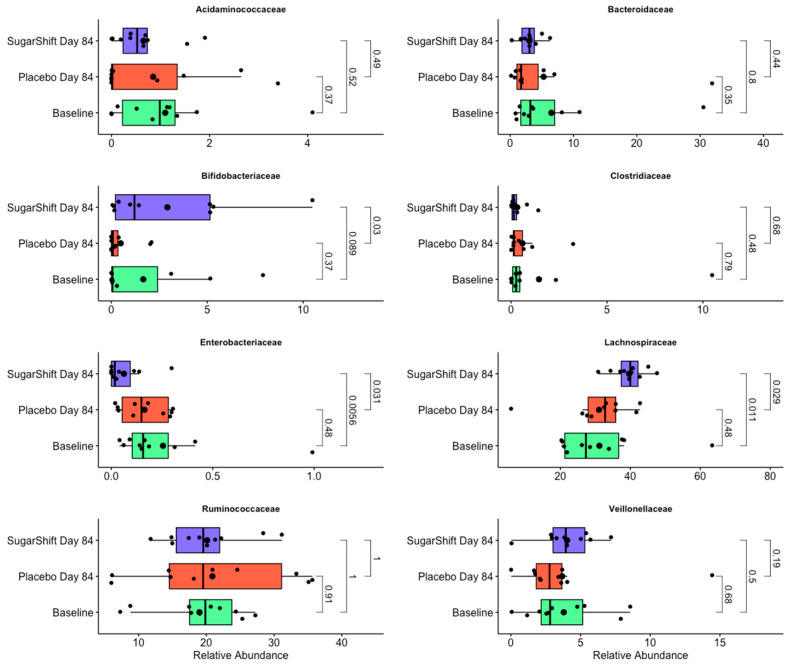
Comparison of eight representative families in the gut microbiome between the baseline, placebo, and Sugar Shift cohorts at the end of the study (Day 84). The boxplots show the relative abundances of four representative bacterial families in metagenomes derived from shotgun sequencing data, comparing the baseline, placebo, and Sugar Shift-treated cohorts at Day 84. The boxes represent the interquartile range (IQR), the horizontal line within the boxes indicates the median, and the whiskers extend to the maximum and minimum values within 1.5 × IQR. Smaller black dots represent individual samples (jittered for visualization), and larger black dots indicate the mean relative abundance for each group. Statistical significance was assessed using the Kruskal–Wallis test. Boxplots were generated using the ggpubr [56] package in R software.

**Figure 5 microorganisms-12-02525-f005:**
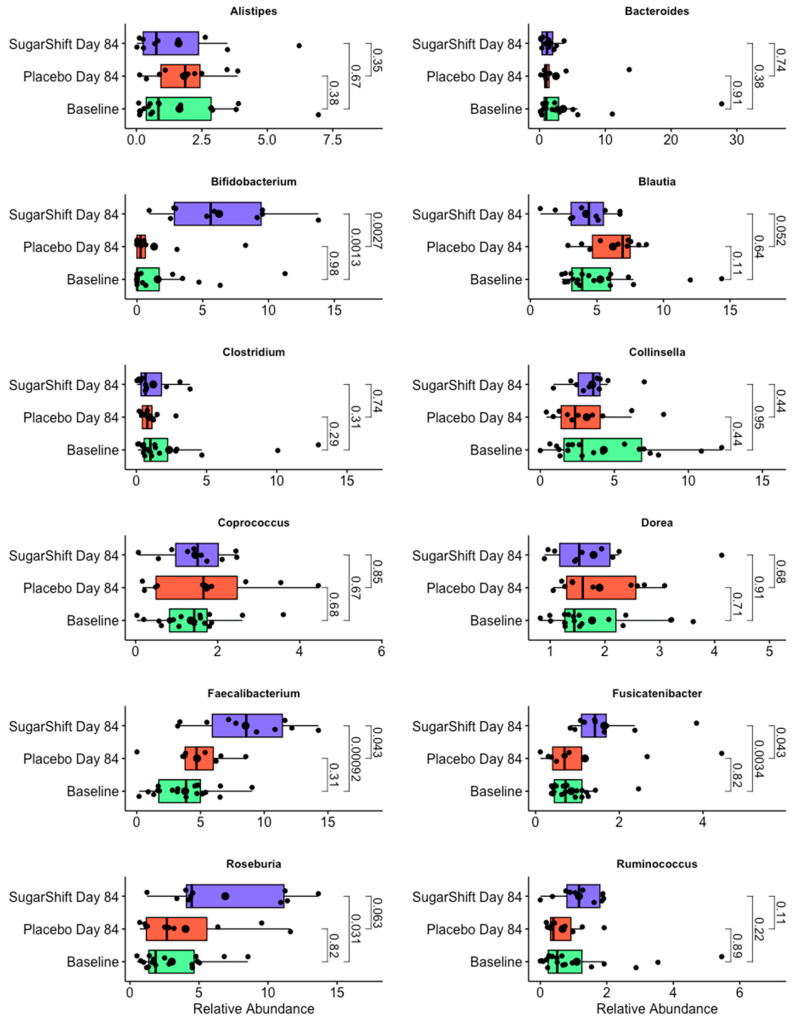
Changes in the relative abundance of key genera between the baseline, placebo, and Sugar Shift-treated cohorts. The boxplots show the relative abundances of four representative bacterial families in metagenomes derived from shotgun sequencing data, comparing the baseline, placebo, and Sugar Shift-treated cohorts at Day 84. The boxes represent the interquartile range (IQR), the horizontal line within the boxes indicates the median, and the whiskers extend to the maximum and minimum values within 1.5 × IQR. Smaller black dots represent individual samples (jittered for visualization), and larger black dots indicate the mean relative abundance for each group. Statistical significance was assessed using the Kruskal–Wallis test. The boxplots were generated using the ggpubr [56] package in R software.

**Figure 6 microorganisms-12-02525-f006:**
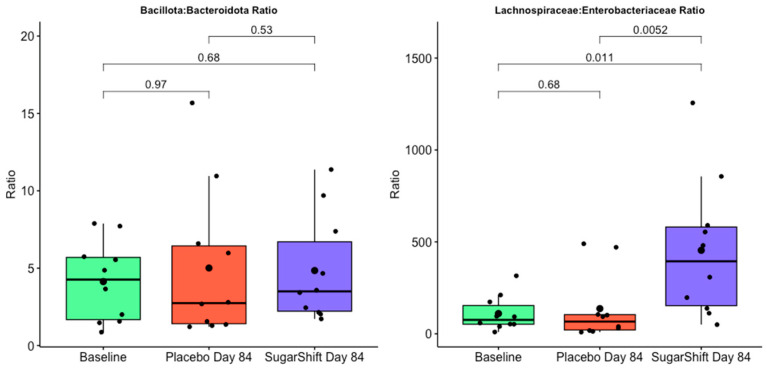
Comparative analysis of the Bacillota:Bacteroidota ratio and the *Lachnospiraceae:Enterobacteriaceae* ratio across baseline, placebo, and treated groups. The boxplots represent the distribution of RA for each genus, with smaller black dots indicating individual samples (jittered for visualization) and larger black dots denoting the mean RA for each group. Statistical significance was assessed using the Kruskal–Wallis test. The boxplots were generated using the ggpubr package in R software.

**Table 1 microorganisms-12-02525-t001:** Distribution of patients according to demographics, nutritional assessment, and kind of treatment.

Demographic Variables		SS Cohort (*n* = 30) ^a^		Placebo (*n* = 27) ^a^		*p*-Value
		No.	%	No.	%	
Sex	Female	18	60.0	14	51.9	0.725 ^b^
	Male	12	40.0	13	48.1	
Age	Median ± SD	56.3 ± 6.7		53.2 ± 7.6		0.120 ^c^
Nutritional Assessment	Normal weight	3	10.0	5	18.5	0.722 ^a^
	Overweight	16	53.3	8	29.6	
	Obesity	11	36.7	14	51.9	
Kind of Treatment	Diet	3	10.0	2	7.4	0.549 ^a^
	Diet plus oral hypoglycemic agents	19	63.3	15	55.6	
	Insulin	0	0.0	2	7.4	
	Combined treatment	8	26.7	8	29.6	

^a^ The sample size (*n* = 32) was calculated to achieve a statistical power of 0.8 (80%) at a significance level of 0.05, based on the expected effect size and variability; ^b^ Friedman test [24]; ^c^ Wilcoxon signed-rank test [25].

**Table 2 microorganisms-12-02525-t002:** Overrepresented orthologs, corresponding genes, and enzymes in the gut microbiome following treatment with Sugar Shift.

Ortholog	Gene	Enzyme	Baseline	Sugar Shift Day 84
			Mean ± SD	Mean ± SD
LPS Biosynthesis Genes				
K02847	*waaL/kdsB*	3-deoxy-manno-octulosonate cytidylyltransferase [EC:2.7.7.38]	59.45 ± 21.55	36.54 ± 5.71
			*p* = 0.006 *	
K02848	*waaP/kdsC*	3-deoxy-manno-octulosonate 8-phosphate phosphatase [EC:3.1.3.45]	13.04 ± 6.31	3.93 ± 1.71
			*p* = 0.0008	
K03760	*eptA/kdsA*	3-deoxy-manno-octulosonate 8-phosphate synthase [EC:2.5.1.55]	99.61 ± 49.02	34.32 ± 5.65
			*p* = 0.005	
K00677	*lpxA*	UDP-N-acetylglucosamine acyltransferase [EC:2.3.1.129]	694.20 ± 304.15	439.13 ± 142.52
			*p* = 0.005	
K00748	*lpxB*	lipid-A-disaccharide synthase [EC:2.4.1.182]	895.65 ± 421.56	470.14 ± 152.40
			*p* = 0.0008	
K02535	*lpxC*	UDP-3-O-[3-hydroxymyristoyl] N-acetylglucosamine deacetylase [EC:3.5.1.108]	1009.99 ± 459.61	400.88 ± 152.02
			*p* = 0.0006	
K02536	*lpxD*	UDP-3-O-[3-hydroxymyristoyl] glucosamine N-acyltransferase [EC:2.3.1.191]	778.59 ± 276.12	371.92 ± 115.31
			*p* = 0.0007	
K03269	*lpxH*	UDP-2,3-diacylglucosamine hydrolase [EC:3.6.1.54]	719.29 ± 233.54	279.04 ± 126.52
			*p* = 0.0005	
K00912	*lphK*	tetraacyldisaccharide 4′-kinase [EC:2.7.1.130]	858.77 ± 272.11	425.71 ± 129.76
			*p* = 0.0003	
K02515	*lphL*	3-deoxy-manno-octulosonate 8-phosphate phosphatase [EC:3.1.3.45]	869.92 ± 240.07	446.64 ± 175.28
			*p* = 0.0005	
SCFA Biosynthesis Genes				
K00925	*ackA*	Acetate kinase [EC:2.7.2.1]	1592.2 ± 969.99	3455.7 ± 1889.65
			*p* = 0.008	
K00175	*crt*	Enoyl-CoA hydratase [EC:4.2.1.17]	365.3 ± 380.37	652.5 ± 396.11
			*p* = 0.037	
K00823	*gabT*	4-aminobutyrate aminotransferase/(S)-3-amino-2-methylpropionate transaminase/5-aminovalerate transaminase [EC:2.6.1.19 2.6.1.22 2.6.1.48]	164.1 ± 133.71	292.3 ± 162.1
			*p* = 0.008	
K18566	*frdA*	NADH-dependent fumarate reductase subunit A [EC:1.3.1.6]	703.1 ± 642.5	1406.3 ± 297.61
			*p* = 0.016	

* Statistical values were calculated using the *t*-test: paired two-sample for means [54].

**Table 3 microorganisms-12-02525-t003:** Summary of the discriminatory biomarkers identified through an LEfSe analysis of the 16S metagenome, normalized for gene copy number.

Taxon Name	*p*-Value	Max Group	COHORT		
			Baseline	Sugar Shift Day 84	Placebo Day 84
*Bacteroides eggerthii*	0.01974	Baseline	1.52822	0.13288	0.00000
CP017245_s	0.03314	Sugar Shift Day 84	0.00000	0.85761	0.00000
PAC001135_s	0.04082	Sugar Shift Day 84	0.10157	0.38603	0.04486
LLKB_g	0.04183	Sugar Shift Day 84	0.14974	0.37675	0.13256
PAC001206_s	0.04590	Sugar Shift Day 84	0.17914	0.18373	0.00000
*Oscillibacter*_uc	0.03216	Baseline	0.22961	0.22653	0.05031
PAC001233_s	0.02572	Sugar Shift Day 84	0.02719	0.20221	0.04486
*Alistipes finegoldii*	0.03909	Sugar Shift Day 84	0.06623	0.12565	0.00729
PAC001036_s	0.03175	Sugar Shift Day 84	0.07542	0.10291	0.00000
*Lactococcus garvieae* gp.	0.01795	Baseline	0.07653	0.00000	0.00000
*Bacteroides xylanisolvens*	0.02666	Baseline	0.06002	0.00745	0.04993
PAC001263_s	0.02927	Sugar Shift Day 84	0.00485	0.03417	0.00000
PAC001458_s	0.03314	Placebo Day 84	0.00000	0.00000	0.02679
PAC001458_g	0.03314	Placebo Day 84	0.00000	0.00000	0.02679

## Data Availability

The metagenomic reads generated from both the 16S rRNA and shotgun sequencing analyses will be deposited in the Sequence Read Archive (SRA) database under BioProject ID: PRJNA1192540.

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
