# Peer review of "Evaluating the Effects of Sugar Shift® Symbiotic on Microbiome Composition and LPS Regulation: A Double-Blind, Placebo-Controlled Study"

_microorganisms, 2024, doi:10.3390/microorganisms12122525_

Round 1
Reviewer 1 Report
Comments and Suggestions for Authors
I have read with great interest the manuscript of Garcia et al titled "Evaluating the Effects of Sugar Shift™ Symbiotic on Microbiome Composition and LPS Regulation: A Double-Blind, Placebo-Controlled Study"
Microbiome-based therapeutics for type 2 diabetes are very interesting to the scientific community.
The manuscript is very well written, however there are some changes to be addressed.
Major comments
Re-write the abstract by defining the purpose of the study in the first lines. State what is known and what is not known regarding your study hypothesis. At the end of your abstract, underline the future perspectives and possible application of your results.
Introduction
The general information regarding the importance of some specific phyla in the microbiota is not comprehensive. The most important phyla Verrucomicrobia represented by Akkermansia muciniphila is not even discussed, although is known to play an important role in metabolic diseases, including type 2 diabetes. There is a lot of information about BiotiQuest, but it would be of interest to discuss why this product is special, and why were those strains chosen related to their importance in metabolism. Try to condense the information regarding your hypotheses and discuss it more in the Discussion section.
Methods
Line 120 - What were the inclusion and exclusion criteria?
Results
It would have been beneficial to see more about the diversity. How did you determine the Shannon index and how is this related to Figure 3 it is not clear. How many phyla did you analyze, and what were the classes of bacteria in each phyla?
Discussion
Were the patients subjected to dietary changes during the study? were they taking any other medication for chronic diseases?
Funding - why was your study funded by a Parkinson’s disease research Foundation?
Minor comments
The manuscript is full of typing errors. Remove Title from the title of the manuscript.
For bacteria please italicize family, genus, species, and variety or subspecies, through the manuscript
Author Response
Reviewer 1.
COMMENT: Re-write the abstract by defining the purpose of the study in the first lines. State what is known and what is not known regarding your study hypothesis. At the end of your abstract, underline the future perspectives and possible application of your results.
RESPONSE: Thank you for pointing this out. We have rewritten the Abstract to include your comments with the 200-word constraint for the Abstract. The revised text has been highlighted in blue. Please, see below:
(1) Background: This study evaluated the effects of BiotiQuest Sugar Shift™, a novel probiotic formulation, for its impact on gut microbiome composition and metabolic health in type 2 diabetes mellitus (T2D). T2D is characterized by chronic inflammation and gut microbiome imbalances, yet the therapeutic potential of targeted probiotics remains underexplored. (2) Methods: In a 12-week randomized, double-blind, placebo-controlled trial, 64 adults with T2D received either Sugar Shift or placebo capsules twice daily. Each dose provided 18 billion CFU of eight GRAS-certified bacterial strains and prebiotics. Clinical samples were analyzed for metabolic markers, and microbiome changes were assessed using 16S rRNA sequencing and metagenomics. (3) Results: Sugar Shift significantly reduced serum lipopolysaccharide (LPS) levels, improved insulin sensitivity (lower HOMA-IR scores), and increased short-chain fatty acid (SCFA)-producing genera, including Bifidobacterium, Faecalibacterium, Fusicatenibacter, and Roseburia. Pro-inflammatory taxa like Enterobacteriaceae decreased, with reduced LPS biosynthesis genes and increased SCFA production genes. The Lachnospiraceae:Enterobacteriaceae ratio emerged as a biomarker of reduced inflammation. (4) Conclusions: These findings demonstrate Sugar Shift’s potential to restore gut homeostasis, reduce inflammation, and improve metabolic health in T2D. Further studies are warranted to explore its long-term efficacy and broader application in metabolic disease management.
COMMENT: The general information regarding the importance of some specific phyla in the microbiota is not comprehensive. The most important phyla Verrucomicrobia represented by Akkermansia muciniphila is not even discussed, although is known to play an important role in metabolic diseases, including type 2 diabetes. There is a lot of information about BiotiQuest, but it would be of interest to discuss why this product is special, and why were those strains chosen related to their importance in metabolism. Try to condense the information regarding your hypotheses and discuss it more in the Discussion section.
RESPONSE: Thank you for your insightful comments. We fully agree and have revised the introduction to address your concerns. The updated sections are highlighted in blue text. You are correct that recent studies have emphasized the importance of the phylum Verrucomicrobia, particularly Akkermansia muciniphila, in preventing type 2 diabetes and obesity in animal models (Liu E., Ji X., Zhou K. Akkermansia muciniphila for the Prevention of Type 2 Diabetes and Obesity: A Meta-Analysis of Animal Studies. Nutrients. 2024; 16(20):3440. https://doi.org/10.3390/nu16203440) and its role in mitigating insulin resistance (Zeng Z., Chen M., Liu Y., et al. (2024) Role of Akkermansia muciniphila in insulin resistance. Journal of Gastroenterology and Hepatology. doi: https://doi.org/10.1111/jgh.16747). However, in our microbiome analysis, we did not observe an increase in Akkermansia, which may be attributed to the specific design of the synbiotic formulation. Given the length constraints of our article, we have chosen to focus on the phyla and microbial shifts directly detected in our study.
COMMENT: There is a lot of information about BiotiQuest, but it would be of interest to discuss why this product is special, and why were those strains chosen related to their importance in metabolism. Try to condense the information regarding your hypotheses and discuss it more in the Discussion section.
RESPONSE: Thank for pointing this out. We agree and have made changes, as indicated in the above response to summarize the features of BiotiQuest in the introduction and include a more detailed discussion in the corresponding section. These are highlighted in blue text as well.
COMMENT: Line 120 - What were the inclusion and exclusion criteria
RESPONSE: Thank you for pointing out this shortcoming. The following text (in blue) has been added to section 2.2 Study Design: The inclusion criteria for the study consisted of participants aged 30 to 65 years (inclusive), of any gender or skin color, with a clinical diagnosis of T2D and who are able and willing to provide informed written consent. Exclusion criteria included patients with chronic kidney disease, oncoproliferative diseases, pregnant women, and individuals with intellectual or psychological dysfunction that impairs their ability to understand and comply with the study requirements, as determined by the Principal Investigator.
COMMENT: It would have been beneficial to see more about the diversity. How did you determine the Shannon index and how is this related to Figure 3 it is not clear. How many phyla did you analyze, and what were the classes of bacteria in each Phylum?
RESPONSE: Thank you for your insightful comments and the opportunity to clarify our methods and findings. The Shannon index was calculated using the vegan package in R and based on the relative abundance data of bacterial taxa identified. Our analysis included 23 phyla and 56 classes of bacteria. To improve clarity, we will include a detailed description of the Shannon index calculation in the Methods section. Regarding the relationship between the Shannon index and Figure 3, the figure represents alpha diversity metrics across treatment groups and time points using three different metrics. We will revise the figure legend and provide additional details. Here is the revised legend for figure 3.
“Figure 3. Boxplots with jitter representing the ACE, Chao1, and Shannon diversity indices for the Baseline, Placebo (Day 84), and Sugar Shift-treated (Day 84) groups. Data visualizations were generated using the ggpubr package in R. Statistical comparisons were conducted using the Kruskal-Wallis test to evaluate differences in species richness and diversity between groups. The Shannon index was calculated using the vegan package in R, based on the relative abundance data of bacterial taxa identified, which included 23 phyla and 56 classes of bacteria.”
To clarify the relationship with Figure 3, it represents three different alpha diversity metrics across treatment groups and time points
We hope these additions will address your concerns and provide greater clarity to our work.
COMMENT: Were the patients subjected to dietary changes during the study?
RESPONSE: No, the patients ate their usual diet.
COMMENT: Were they taking any other medication for chronic diseases?
RESPONSE: The patients continued their usual diet throughout the study. It is important to note that most diabetic patients also suffer from hypertension; therefore, in addition to their diabetes treatments, they were receiving antihypertensive medications. However, these treatments were not considered in the current analysis.
COMMENT: Why was your study funded by a Parkinson’s disease research Foundation?
RESPONSE: As indicated, the study on type 2 diabetes (T2D) was funded by The BioCollective Foundation, a nonprofit organization dedicated to supporting research on Parkinson’s Disease and other microbiome-related conditions. The initial development of Sugar Shift was specifically aimed at addressing symptoms of Parkinson’s Disease, serving as a palliative measure to improve gut health and metabolic function, which are often impaired in individuals with Parkinson’s.
Parkinson’s is sometimes referred to as "type 3 diabetes" due to shared underlying mechanisms, such as insulin resistance, chronic inflammation, and disrupted glucose metabolism in the brain. Given these overlapping pathways, investigating Sugar Shift’s effects on T2D not only provides critical insights into its potential role in managing metabolic health but may also yield broader benefits for conditions like Parkinson’s Disease. Supporting this research aligns with The BioCollective Foundation’s mission to explore innovative solutions for microbiome-related diseases and improve patient outcomes.
Reviewer 2 Report
Comments and Suggestions for Authors
The manuscript under review presents an exploration into the impact of the Sugar Shift™ probiotic formulation on gut microbiome composition and LPS regulation in individuals with Type 2 Diabetes (T2D). Utilizing 16S rRNA sequencing and shotgun metagenomics, the study evaluates taxonomic and functional alterations, particularly in the context of short-chain fatty acid (SCFA) production and lipopolysaccharide (LPS) biosynthesis. The findings suggest that the Sugar Shift™ intervention notably elevated the Lachnospiraceae to Enterobacteriaceae ratio, a potential biomarker for gut health in T2D patients. The study also reports a reduction in inflammatory taxa such as Enterobacteriaceae and an enrichment of SCFA-producing bacteria like Faecalibacterium prausnitzii. Functional analyses indicate improved SCFA biosynthesis and suppressed LPS biosynthetic gene expression, correlating with clinical improvements in serum LPS levels and insulin resistance as measured by the HOMA-IR index.
The manuscript is overall well written and clearly presented, making it a suitable candidate for publication in Microorganisms after addressing minor revisions. Prior to publication, the following points require clarification:
1. In line with Microorganisms' guidelines, the keywords should be limited to ten. The authors should specify the essential keywords of the article.
2. Line 116: As described here, a 12-week clinical study was conducted. Is this short period sufficient to observe the impact of microbiome changes on long-term health?
3. The Discussion section is noted to be excessively lengthy. The authors may concisely compare their findings regarding SCFA and LPS production with those from other studies, focusing on the unique contributions of their research.
4. whether the authors could discuss more details about the potential mechanisms by which the probiotic intervention influences gut microbiome composition and function?
Author Response
COMMENT: In line with Microorganisms' guidelines, the keywords should be limited to ten. The authors should specify the essential keywords of the article.
RESPONSE: You are correct, We will restrict the keywords to the following: Diabetes, Microbiome, Probiotics, Lipopolysaccharides, SCFA, Clinical Trials, SugarShift
COMMENT: Line 116: As described here, a 12-week clinical study was conducted. Is this short period sufficient to observe the impact of microbiome changes on long-term health?
RESPONSE: You are correct that 12 weeks is a relatively short period, especially if one looks for significant changes in the Hb A1c levels. However, a meta-analysis examining the average duration of probiotic administration across 33 clinical trials in patients with diabetes mellitus reported a mean duration of 2.861 months. The duration of probiotic treatment ranged from 1 to 6 months, with 18 studies administering probiotics for ≥3 months and 15 studies for <3 months (Paquette, S.; Thomas, S.C.; Venkataraman, K.; Appanna, V.D.; Tharmalingam, S. The Effects of Oral Probiotics on Type 2 Diabetes Mellitus (T2DM): A Clinical Trial Systematic Literature Review. Nutrients 2023, 15, 4690).
This study represents the first clinical trial in which we evaluated the symbiotic formulation specifically designed to modulate clinical parameters in type 2 diabetes (T2D). Based on our results, we now understand that during the trial period, stabilization of clinical parameters in T2D occurred, making this an opportune time to induce microbiome turnover. The long-term impact of microbiome changes will depend significantly on other environmental factors, such as the individuals' nutritional habits and engagement in physical exercise. Continuous monitoring of LPS levels over a period of at least one to two years, along with evaluations of sustained compensation in clinical parameters, would be essential to determine the duration of the microbiome's effects. Your observation is an excellent idea for a novel clinical trial and for the follow-up of our patients, providing an opportunity to further understand and validate these findings.
COMMENT: The Discussion section is noted to be excessively lengthy.
RESPONSE: Yes, you are correct; we have shortened the discussion section.
COMMENT: The authors may concisely compare their findings regarding SCFA and LPS production with those from other studies, focusing on the unique contributions of their research
RESPONSE: We rewrote section 4.1 Clinical parameters (in blue text) to address the reviewer’s concerns.
COMMENT: whether the authors could discuss more details about the potential mechanisms by which the probiotic intervention influences gut microbiome composition and function?
RESPONSE: We have addressed the reviewer's suggestion by including a dedicated section, 4.7. Mechanisms of Gut Microbiome Modulation by the Probiotic Sugar Shift in Type 2 Diabetes Management, in the discussion. This section discusses how the probiotic Sugar Shift (SS) modulates gut microbiome composition and function to support better metabolic outcomes in type 2 diabetes (T2D) management.
Reviewer 3 Report
Comments and Suggestions for Authors
The authors employed microbiome analysis techniques to compare the efficacy of a symbiotic product compared to placebo, in a double-blind, placebo-controlled study. The findings are very interesting and relevant to the clinical practice. The authors concluded that this product intervention led to a healthier gut microbiome and reduced inflammation while the inflammatory bacteria were reduced.
Also, the authors documented improvements in metabolic markers of insulin resistance, concluding that Sugar Shift may modulate the gut microbiota to improve metabolic health in type 2 diabetes patients.
The manuscript is well written and has a high scientific sound.
I only have some minor suggestions:
Please correct the title by deleting the word title and ” from the end
The first author has an affiliation that is not listed. Update or delete a
The second keyword has an extra space.
Abbreviations are usually defined at the first use in the abstract as well as in the main text.
Author Response
Reviewer 3
COMMENT: Please correct the title by deleting the word title an ” from the end
RESPONSE: Thank you for pointing that error, it has been corrected in the revised manuscript.
COMMENT: The first author has an affiliation that is not listed. Update or delete a
RESPONSE: Thank you for pointing that error, it has been corrected in the revised manuscript.
COMMENT: The second keyword has an extra space.
RESPONSE: Thank you for pointing that out. We have revised the list of keywords to comply with the journal’ guidelines.